# Language Models as Zero-shot Lossless Gradient Compressors: Towards General Neural Parameter Prior Models

**Hui-Po Wang    Mario Fritz**
CISPA Helmholtz Center for Information Security
Germany
{hui.wang, fritz}@cispa.de

## Abstract

Despite the widespread use of statistical prior models in various fields, such models for neural network gradients have long been overlooked. The inherent challenge stems from their high-dimensional structures and complex interdependencies, which complicate effective modeling. In this work, we demonstrate the potential of large language models (LLMs) to act as gradient priors in a zero-shot setting. We examine the property by considering lossless gradient compression – a critical application in distributed learning – that depends heavily on precise probability modeling. To achieve this, we introduce LM-GC, a novel method that integrates LLMs with arithmetic coding. Our technique converts plain gradients into text-like formats, enhancing token efficiency by up to 38 times compared to their plain representations. We ensure that this data conversion maintains a close alignment with the structure of plain gradients and the symbols commonly recognized by LLMs. Our experiments indicate that LM-GC surpasses existing state-of-the-art lossless compression methods, improving compression rates by 10% up to 17.2% across various datasets and architectures. Additionally, our approach shows promising compatibility with lossy compression techniques such as quantization and sparsification. These findings highlight the significant potential of LLMs as a model for effectively handling gradients. Code is available at `https://github.com/hui-po-wang/LM-GC`.

## 1   Introduction

Statistical prior models have been applied successfully in various fields, including image denoising and super-resolution [Ulyanov et al., 2018, Gandelsman et al., 2019], vision task adaptation [Wang et al., 2021], and low-resource language tasks [Baziotis et al., 2020, Brown et al., 2020]. However, their use in modeling neural network gradients has been largely neglected. The potential reasons for this oversight might include (1) the high-dimensional nature of gradients, which makes them less intuitive to analyze; (2) the difficulty of collecting representative gradient data; and (3) the significant challenge of ensuring generalizability to unseen data, given the substantial effort required.

Instead of developing a model from scratch, this work investigates the potential of leveraging pre-trained large-scale language models (LLMs) as gradient priors in a zero-shot setting. We explore this potential through the lens of lossless gradient compression, a vital application in federated and distributed learning environments. The success of this compression heavily depends on precise probability modeling. An effective statistical model can significantly improve compression efficiency, whereas an inaccurate model may lead to poorer compression outcomes and could even increase the data size post-compression.

38th Conference on Neural Information Processing Systems (NeurIPS 2024).

To address this, we introduce LM-GC, an innovative coding scheme for gradient compression that integrates pre-trained large language models (LLMs) with arithmetic coding. Our method involves transforming gradients into text-like formats that are easier for LLMs to reason. Specifically, we convert the raw bit data of floating points into hexadecimal numbers and incorporate separators, such as spaces, to clarify the concept of floating points for LLMs. This serialized text is then processed by pre-trained tokenizers and LLMs to determine the probability of each token, which is subsequently utilized in arithmetic coding. Empirical evidence supports that these design choices significantly enhance gradient modeling and, consequently, compression efficiency.

Overall, our contributions are summarized below.

- We introduce LM-GC, a novel coding scheme that integrates LLMs with arithmetic coding. This method utilizes LLMs as powerful prior models for gradients, setting a new benchmark in state-of-the-art lossless gradient compression.

- LM-GC demonstrates that transforming raw gradients into formats that LLMs can understand significantly impacts their reasoning capabilities and token efficiency. Empirical evidence indicates that this approach can affect compression rates up to 70% with recognizable symbols and 40% with proper separators. These findings underscore the critical role of effective conversion in enhancing compression performance.

- Experimental results demonstrate that LM-GC significantly surpasses existing baselines, including PNG, FLAC, LZMA, GZIP, and FPZIP, by 10% to 17.2% across various architectures and datasets. Additionally, our approach complements existing lossy compression methods such as quantization and sparsification, paving the way for advanced gradient compression techniques.

## 2    Related work

**Large-scale language models.**    Language models aim to model the relation between texts. This problem has been extensively studied in recent decades via various approaches such as statistical models [Jelinek, 1998] and recurrent neural networks [Hochreiter and Schmidhuber, 1997]. Recently, the emergence of transformer-based models [Vaswani et al., 2017] along with large-scale text corpora has revolutionized the entire field, driving research into large-scale language models (LLMs). Models, such as those from GPT [Achiam et al., 2023, Brown et al., 2020] and LLAMA [Touvron et al., 2023, Zhang et al., 2024, Geng and Liu, 2023] families, are capable of solving diverse tasks in natural languages and demonstrate incredible generalizability toward unseen novel tasks, even across modalities [Mirchandani et al., 2023, Gruver et al., 2024]. Notably, recent work by Deletang et al. [2024] also explores the use of language models as general compressors. Our goal is to investigate the potential of LLMs as a strong prior specifically for gradients. Additionally, we offer practical considerations for handling floating-point data when structures exist within the data to be compressed.

In this work, we demonstrate for the first time that LLMs can understand the structure of network gradients, accurately modeling their probability distribution in a fully zero-shot manner. We verify our finding by taking LLMs as priors for arithmetic coding, yielding state-of-the-art lossless gradient compression under various settings.

**Deep generative priors.**    An ongoing research direction beyond traditional statistical modeling is learning a deep generative model from massive data and leveraging the model as a "deep" prior. The concept has been widely considered in many applications, such as image denoising and super-resolution [Ulyanov et al., 2018, Gandelsman et al., 2019], vision task adaptation [Wang et al., 2021, Chang et al., 2019], and low-resource language tasks [Baziotis et al., 2020, Brown et al., 2020]. Although strong priors can facilitate various downstream applications, training such models for gradients can be costly and challenging due to their high dimensionality. Additionally, the generalizability of these models is often a concern and may be limited to specific types of networks [Ha et al., 2016, Wang et al., 2024b]. Instead of training a model from scratch, our work explores the potential of using off-the-shelf LLMs as strong priors over gradients. This will minimize the cost of training deep prior models and may inspire applications like gradient denoising and anomaly detection.

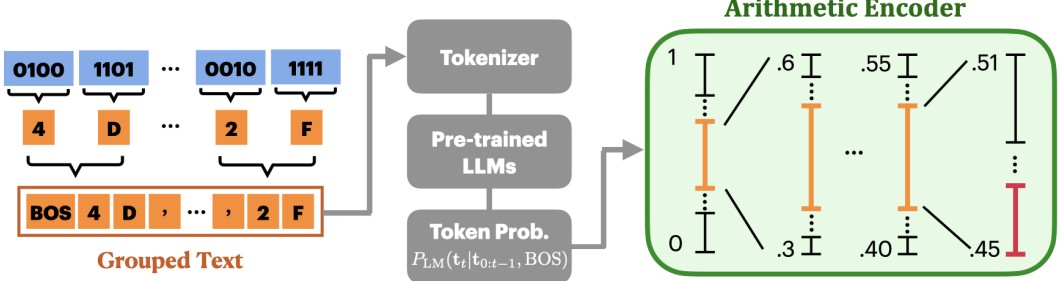

Figure 1: Overview of LM-GC. Our method initially converts every 4 bits into hexadecimal numbers and groups them with separators in between, e.g., commas in the figure. The grouped text is then input to a pre-trained, frozen tokenizer and LLM to produce the probability of each token. These probabilities are used for arithmetic encoding, where a line segment between 0 and 1 is repeatedly split according to the token probability until reaching a predefined maximum length. Any number from that region (e.g., the midpoint) can accurately represent the original data. We provide an example of how arithmetic coding works in Sec. 3.

**Gradient compression.** Gradient compression is a crucial technique, particularly in federated and distributed learning, where communication costs serve as the main bottleneck for scalability. Existing efforts have extensively studied lossy compression, which trades information precision for compression efficiency. For example, quantization [He et al., 2020, Alistarh et al., 2017, Bernstein et al., 2018] replaces floating points with fewer bits, while sparsification [Wangni et al., 2018, Alistarh et al., 2018] transmits only a subset of the original gradients. Other approaches explore novel optimization strategies such as progressive learning [Wang et al., 2022] and communicating synthetic images [Wang et al., 2024a, Xiong et al., 2023].

In contrast, lossless compression, which allows compressed data to be perfectly reconstructed without sacrificing information, is rarely investigated in the field of gradient compression nowadays. The challenge lies in developing a better statistical model for gradients. In this work, we demonstrate that LLMs can model the probability distribution of gradients in a zero-shot setting, Building on this finding, our method combines LLM-based modeling with arithmetic encoding and outperforms existing baselines such as PNG [Boutell, 1997], FLAC [Coalson, 2008], LZMA [Pavlov, 2019], GZIP [Deutsch, 1996], and FPZIP [Lindstrom and Isenburg, 2006], which are designed for modalities other than gradients. By integrating our approach with existing lossy compression techniques, we may pave the way for more advanced gradient compression schemes.

## 3 Background

In this work, we aim to explore the potential of using LLMs as prior for gradients and leverage lossless compression as an examination task. We review the essential background knowledge below.

**Lossless compression.** The fundamental principle of lossless compression is to reduce the size of data while ensuring it can be fully reconstructed. This is typically achieved by eliminating the statistical redundancy inherent in the data. Given a sequence of symbols $s_{0:N} \in \mathcal{S}$ drawn from a probability distribution $P_{\mathcal{S}}$, the objective is to devise a compression function $g : \mathcal{S} \to \mathcal{C}$. This function maps the original data $s$ to a unique (decodable) binary code $c$, ensuring that the length of $c$, denoted by $\ell(c)$, is less than or equal to the length of $s$, $\ell(s)$. The source coding theorem [Shannon, 2001] states that the expected minimum length of a coded message $c$ cannot be shorter than the Shannon entropy of the original data, denoted as $\ell(s) \geq H(\mathcal{S})$. Here, $H(\mathcal{S}) := \mathbb{E}_{s \sim P_{\mathcal{S}}}[-\log_2 P_{\mathcal{S}}(s)]$ represents the entropy. This implies that any compression resulting in a length shorter than $H(\mathcal{S})$ necessarily involves loss of information, preventing perfect reconstruction of the original data.

**Arithmetic coding.** As a means to achieve lossless compression, arithmetic coding [Rissanen and Langdon, 1979] provides a nearly optimal message length $H(\mathcal{S}) \leq \ell(c) \leq H(\mathcal{S}) + 2/\ell(s)$

on average [Sayood, 2017]. To implement it, one must employ a statistical probability model of the data, denoted $P_{AC}$. Ideally, this model $P_{AC}$ should closely mirror the true distribution $P_S$. *The closer these distributions are, the more effective the compression performance will be.* Conversely, significant deviations can result in a compressed data length $\ell(c)$ that exceeds the original data length $\ell(s)$. Notably, Most existing methods, such as CABAC, incorporate adaptive priors, meaning the probability $P_{AC}$ adapts based on data context. However, as we will demonstrate in Sec. 5, these methods are not optimized for gradients and are thus outperformed by our zero-shot LLM prior.

The arithmetic encoder begins with an interval [0, 1) between 0 and 1. For each input symbol $s$, the interval is subdivided according to the probability $P_{AC}(s)$. The corresponding interval is then selected as the new interval. This process continues until the entire input stream is finished or reaches the maximum length. Any number existing in the final interval suffices to represent the compressed data. Similarly, the decoder takes the encoded output as input and can perfectly reconstruct the data by repeatedly looking up the intervals.

We provide an example illustrating how arithmetic coding works given a fixed statistical prior below.

**Example of arithmetic coding.** Consider a message consisting of only two symbols, A and B, where A occurs with probability $P_{AC}(A) = 0.8$ and B with $P_{AC}(B) = 0.2$. The encoding interval gets subdivided into a larger part (0 to 0.8) for A and a smaller part (0.8 to 1) for B. If the message is "AAAB", the interval narrows from [0, 1] to [0, 0.8], then [0, 0.64], then [0, 0.512], and finally [0.4096, 0.512] after processing the B. Any number (typically the midpoint for simplicity) within the final interval can represent the entire sequence. This number is then converted into a binary code, which is the compressed output. The final result is $(0.4608)_{10} \rightarrow (0.01110101)_2$, which takes only 8 bits compared to 4 bytes of storage for the ASCII format.

**Language models.** Language models are designed to model the relation between text symbols. Given a text stream, $S = \{s_i\}^N$, consisting of N symbols, modern language models typically begin with a tokenization process $f : S \rightarrow T$ (e.g., Byte-Pair Encoding [Sennrich et al., 2015]) that maps the entire stream to a set of K tokens $T = \{t\}^K$. Then, the model predicts the probability as follows.

$$p(t) = p(t_1, \ldots, t_K) = \prod_{k=1}^{K} p(t_k | \texttt{BOS}, t_{<k}), \tag{1}$$

where the BOS token denotes a special token indicating the beginning of the sentence.

## 4  LM-GC

We introduce LM-GC, a method that integrates arithmetic coding with pre-trained large language models (LLMs) to address the lack of gradient-specific priors in arithmetic coding. It's important to note that LLMs are originally trained on extensive text corpora and do not encounter gradients or model parameters during this training. A significant challenge is enabling LLMs to comprehend the structure of gradients. Our method involves two main steps: serialization and compression. In serialization, we convert the 32-bit floating points of gradients into a format understandable by LLMs, which we call grouped text. This text is then fed into the LLMs, which predict the probability of each token in an autoregressive manner and thus accomplish compression using arithmetic coding.

**Serialization.** We first note that gradients are represented as 32-bit floating points, with values ranging from $-3.40282347 \times 10^{+38}$ to $-1.17549435 \times 10^{-38}$. Due to significant variations in their magnitudes and the often ambiguous importance of each gradient element, directly inputting these values into large language models (LLMs) is impractical. LLMs have a fixed token limit, and representing a single gradient in plain form would consume excessive tokens, compromising the context's depth.

To address this, our method, LM-GC, initially divides the floating points into several disjoint 4-bit partitions, which are then encoded into hexadecimal numbers as illustrated in Figure 1. This encoding strategy allows for a token savings of approximately 38 times compared to using plain gradients, particularly under extreme value conditions.

Furthermore, we organize every eight decoded hexadecimal numbers (equivalent to 4 bytes) by inserting a separator between them. This format provides LLMs with a structured representation of how a floating point number is typically presented. Our experiments demonstrate that separators are crucial in effectively modeling gradients, especially gradients derived from sophisticated architectures and datasets.

**Compression.** After serializing the gradients into grouped text $\mathcal{S}$, we process this text through a tokenizer to generate a set of tokens $\mathcal{T}$. These tokens are then fed into a pre-trained large language model (LLM), denoted by $\mathcal{M}$, which predicts the probability of each token as follows:

$$P_{\text{LM}}(\mathcal{T}) := \prod_{k=1}^{K} p(t_k | \text{BOS}, t_{<k}). \tag{2}$$

This equation indicates that the LLM sequentially predicts the probability of the next token, starting with the BOS (beginning of sequence) token, which is used to calculate the probability of the first token, $P_{\text{LM}}(t_1) = \mathcal{M}(\text{BOS})$. The BOS token serves primarily as a contextual cue and is not included in the compression.

During compression, $P_{\text{LM}}(\mathcal{T})$ acts as the statistical model, $P_{\text{AC}}$, for arithmetic coding. In the decompression phase, the process begins with the BOS token, retrieving $P_{\text{LM}}(t_1)$ to decode the first token. This decoding cycle continues until the maximum window size of the LLM is reached.

# 5 Experiments

## 5.1 Setup

**Datasets and models.** Our experiments consider three types of LLMs as the compressor, including Tinyllama 1.1B [Zhang et al., 2024], Openllama 3B [Geng and Liu, 2023], and LLAMA 2 7B [Touvron et al., 2023], ranging from a smaller to medium model size. All models can accept up to 4096 tokens. Unless stated otherwise, we use a context window size of 2048 by default. To ensure generalizability, we conduct experiments on four model architectures, including a three-layer convolution net (ConvNet), VGG-16 [Simonyan and Zisserman, 2015], ResNet [He et al., 2016], and vision transformer (ViT) [Dosovitskiy et al., 2021]. The models are trained on three datasets, MNIST [LeCun et al., 2010], CIFAR-10 [Krizhevsky et al., 2009], TinyImageNet [Le and Yang, 2015], under different settings. MNIST is a digit classification task containing 10 digits and 60000 images. On the other hand, CIFAR-10 and TinyImageNet are image classification tasks. CIFAR-10 contains 50000 images of 10 classes, while TinyImageNet contains 100000 images for 200 classes. All images are rescaled to 32 by 32 in the experiments.

**Evaluation protocols.** If a prior describes data well, lossless compression can achieve better efficiency. To test the efficiency of using LLMs as priors, we first train models on different datasets for 200 epochs, collecting gradients every 200 batch steps, resulting in a data pool of approximately 400 checkpoints for compression evaluation. Due to computational time constraints, we sub-sample 10 checkpoints from the pool for the subsequent experiments unless stated otherwise. All the experiments are repeated at least three times, and the standard deviations are reported accordingly. We measure compression efficiency by the compression rates defined as follows.

$$\text{Compression Rate (\%)} = 100 \times \frac{\text{Compressed Data Size}}{\text{Original Data Size}} \tag{3}$$

**Baselines.** We compare our method to state-of-the-art lossless compression techniques that originally targeted different data types. PNG [Boutell, 1997] is one of the most common lossless compression codecs for images. On the other hand, FLAC [Coalson, 2008] is a common audio compression format. Lastly, LZMA [Pavlov, 2019] and GZIP [Deutsch, 1996] are codecs used by 7-zip software and 7z compression format. FPZIP [Lindstrom and Isenburg, 2006] is proposed for scientific floating-point data, particularly suitable for data with up to 4D structures.

| | Traditional codec | | LM-GC (Ours) | | | | | |
|---|---|---|---|---|---|---|---|---|
| | Unchunked | Chunked | ISO | $H_n$ | $H_s$ | $H_c$ | $H_{c+s}$ | $H_{semi}$ |
| PNG | 43.30±1.3 | 49.18±1.1 | | | | | | |
| FLAC | 52.37±0.6 | 50.46±0.6 | | | | | | |
| GZIP | 42.42±0.3 | 47.10±0.4 | | | | | | |
| LZMA | 41.91±0.0 | 47.36±0.1 | | | | | | |
| FPZIP | 41.26±0.8 | 49.27±0.3 | | | | | | |
| Tinyllama 1.1B | | | 117.38±0.0 | 36.30±0.8 | 38.83±0.4 | 38.40±0.6 | 38.46±0.1 | 43.45±0.6 |
| Openllama 3B | | | 71.85±0.2 | 37.07±0.1 | 32.32±0.3 | 34.31±0.6 | 33.07±0.5 | 33.57±0.2 |
| LLAMA 2 7B | | | 109.07±0.2 | 72.10±0.5 | 32.26±0.5 | 32.96±0.3 | **32.21±0.8** | 32.78±0.4 |

Table 1: Gradient compression rate using PNG, FLAC, GZIP, LZMA, FPZIP, and our method with various language models. Our method considers different serializations including iso-8859-1 (ISO), hexadecimal numbers without separators ($H_n$) and with spaces ($H_s$), commas ($H_c$), commas+spaces ($H_{s+c}$), and semicolons ($H_{semi}$) to group every four bytes from the same floating point.

**Implementation.** We implement our method in Pytorch and Huggingface. The checkpoints of pre-trained LLM models are loaded from the Huggingface hub. We adapted the arithmetic coding from Torchac to fit our application. We run our experiments on a cluster with NVIDIA A100 40GB GPUs and AMD EPYC 7402 24-Core Processor. All of the experiments can fit in one single A100.

## 5.2 Compression effectiveness

We first conduct compression experiments on gradients collected from a ConvNet trained on CIFA-10 to show that LLMs can model gradients even without seeing such data during training. Our method considers three LLMs, namely Tinyllama, Openllama, and LLAMA 2, as the priors for arithmetic coding. We also consider 6 types of serialization, including decoding every byte with ISO-8859-1 (ISO), projecting every 4 bits to hexadecimal numbers without separators ($H_n$), and with space ($H_s$), commas ($H_c$), space and commas ($H_{c+s}$), and semicolons ($H_{semi}$) as the separators. These settings outline the importance of serialization and its effect on gradient modeling. We report two settings for the baselines. The first is a *chunked* version, where the compressor sees a chunk of size 512 bytes every time, whereas the other one, namely the *unchunked* version, takes advantage of the pseudo infinitely large context length to yield the best statistical modeling.

Table 1 shows that our LM-GC consistently outperforms baseline codecs when serialization is properly managed. For example, ISO and $H_n$ for LLAMA 2 perform worse than the baselines. In particular, ISO encodes gradients into symbols less familiar to LLMs, yielding up to 70% performance difference compared to settings like $H_s$. The lack of separators may confuse language models, causing performance degradation of 40% on LLAMA 2. These results highlight the crucial role of serialization in aiding LLMs' understanding. Furthermore, compression efficiency increases as the model size grows from 1.1B to 7B, suggesting that more sophisticated models may better understand the relationships between data elements, resulting in more effective compression.

## 5.3 Ablation study

In this section, we provide a series of ablation studies to provide insights into how design choices affect LLMs and the resulting prior models.

**Architectures.** To further understand the generalizability to different architectures and the effect of serialization, we continue with an experiment on different architectures. We extend the experiments to three additional architectures. VGG-16 contains deeper layers compared to ConvNets. ResNet-18 further introduces skip-connections and batch normalization, verifying our LM-GC on common design choices in modern machine learning. Lastly, ViT is built upon transformer blocks, showing that LLMs can reason beyond convolution layers. As shown in Table 2, we first observe that the performance of all methods drops as the models become more complicated, while our method remains the best among the baselines. This finding suggests that our method can better capture complex structures within the gradients. Moreover, we observe that serialization with separators generally

| | Traditional codec | | | | | Ours (Tinyllama 1.1B) | | | |
|---|---|---|---|---|---|---|---|---|---|
| | PNG | FLAC | GZIP | LZMA | FPZIP | $H_n$ | $H_s$ | $H_c$ | $H_{c+s}$ |
| ConvNet | 43.30±1.3 | 52.37±0.6 | 42.42±0.3 | 41.91±0.0 | 41.26±0.75 | **36.30±0.8** | 38.83±0.4 | 38.40±0.6 | 38.46±0.1 |
| VGG16 | 95.61±0.2 | - | 91.91±0.0 | 91.27±0.1 | 89.15±0.17 | 83.23±0.0 | **73.42±0.1** | 75.32±0.2 | 73.97±0.1 |
| ResNet18 | 97.22±0.1 | - | 92.47±0.0 | 91.72±0.1 | 90.72±0.07 | 83.20±0.3 | **73.57±0.1** | 75.55±0.3 | 73.95±0.2 |
| ViT | 94.50±0.4 | - | 89.20±1.2 | 87.98±1.2 | 89.77±0.48 | 78.65±3.3 | **70.83±1.8** | 72.60±2.0 | 71.62±1.7 |

Table 2: Gradient compression (%) for convolution neural networks (ConvNet), VGG-16, ResNet-18, and ViT trained on CIFAR-10.

| | Traditional codec | | | | | | |
|---|---|---|---|---|---|---|---|
| | PNG | FLAC | GZIP | LZMA | FPZIP | LM-GC ($H_s$) | Impr. |
| MNIST | 50.05±4.3 | 55.20±1.7 | 45.05±5.2 | 43.19±1.3 | 44.62±0.6 | **39.38±1.4** | 8.8% |
| CIFAR-10 | 43.30±1.3 | 52.37±0.6 | 42.42±0.3 | 41.91±0.0 | 41.26±0.8 | **38.83±0.4** | 5.9% |
| TinyImageNet | 96.08±0.0 | 107.36±0.0 | 92.18±0.0 | 91.06±0.1 | 86.88±0.1 | **71.90±0.0** | 17.2% |

Table 3: Compression effectiveness on MNIST, CIFAR-10, and TinyImageNet datasets. We use a Tinyllama as the compressor to compress the gradients of ConvNets. The raw data are converted to hexadecimal numbers with spaces as the separator. The improvement (Impr.) over the best baseline highlights the capability of LM-GC in modeling complex gradients.

performs better than the one without separators. It outlines the importance of separators, especially when the data to be compressed becomes more intricate.

**Datasets.** The previous experiment suggests that LM-GC models gradients more accurately than the existing baselines, especially when considering complex structures. We further explore this dimension by considering two additional datasets, MNIST and TinyImageNet. Table 3 presents the result comparing our method with Tinyllama to the baselines. Datasets like TinyImageNet introduce higher compression difficulty due to the complex task. However, LM-GC demonstrates consistently promising performance across all datasets. The improvement over the best baselines (FPZIP) increases as the dataset becomes sophisticated. This finding aligns with the result in Table 2 that our method is generalizable and better at capturing complex structures than the existing codecs that are not optimized for gradient compression.

**Context window size.** LM-GC takes LLMs as prior over gradients. One natural question is whether the LLMs really consider the context and yield accurate probability modeling. Ideally, similar to the traditional codec, if we provide a larger context window, the statistical model should be able to

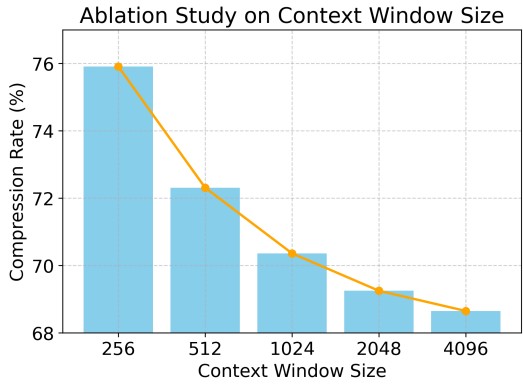

Figure 2: Compression rates of LLAMA 2-7B using context window sizes of 256, 512, 1024, 2048, and 4096. The compression rates improve as the context window increases.

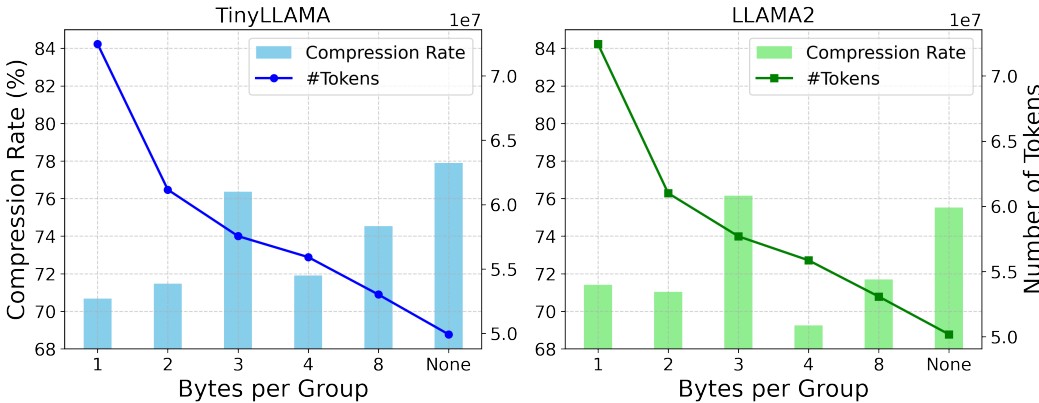

Figure 3: Ablation study on numbers of grouped bytes. We report the compression rates and the number of tokens yielded by different serializations. The settings that closely obey the data format perform better. However, smaller numbers yield higher computation overhead.

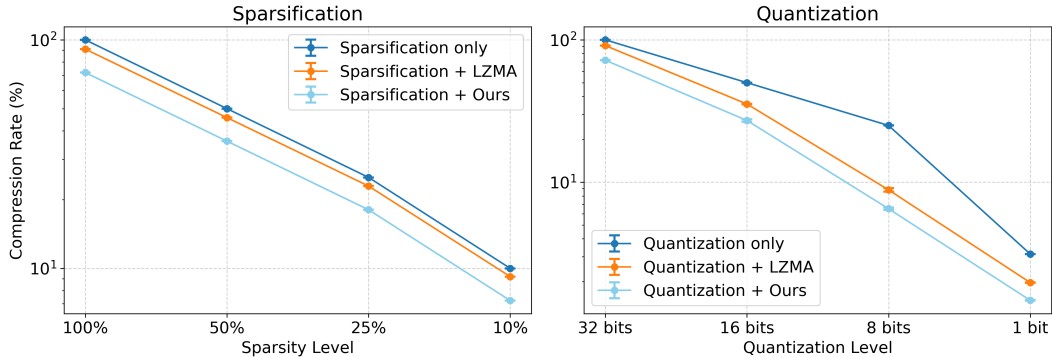

Figure 4: Compatibility analysis with sparsification (left) and quantization (right).

reason from the context and thus result in higher compression efficiency. Instead of using a default context window size of 2048 tokens, we conduct an ablation study in Fig. 2. The result shows that the performance drastically improves when the context window size increases, suggesting that LLMs indeed leverage the context. However, we note that the improvement seems to be saturated at the end. A larger context window also implies higher hardware resource demands, leaving a potential trade-off in practice.

**Byte grouping.** In addition to the decoding schemes analyzed in the previous experiments, we demonstrate that grouping converted text significantly affects performance. Recall that a floating point consists of 1 bit for the sign, 8 bits for the exponent, and 23 bits for the mantissa. Components with the same functionality should be grouped as closely as possible. To verify this hypothesis, we conducted experiments on TinyLLAMA and LLAMA 2 with bytes per group (BPG) set to 1, 2, 3, 4, 8, and none (i.e., no grouping, denoted as $H_n$). The results in Fig. 3 show that BPG set to 1, 2, and 4 (our default setting) perform the best, while BPG equal to 3, which covers three components, and none perform worst. It indicates that serialization should resemble the structure of data to be compressed. Notably, although BPG equal to 1 and 2 performs well on both models, smaller BPG will add more separators and increase the total amount of tokens, introducing the additional computation overhead to the compression.

**Comparison to run-length encoding.** Lastly, we compare our method to run-length encoding, the simplest adaptive compression scheme, as shown in Table 4 in the appendix. The results indicate that although serialization may slightly improve compression rates, run-length encoding is ineffective

for compressing gradients. Combined with the earlier findings, this suggests that simple adaptive methods are unsuitable for handling complex yet structured gradient data.

## 5.4 Compatibility

Gradient compression, a crucial technique in federated learning, typically involves lossy compression methods. We demonstrate that our LM-GC approach is compatible with lossy techniques such as quantization and sparsification. Specifically, we consider linear quantization, which uniformly discretizes value ranges according to the allotted bits. Given a vector $v$ and $n$ bits, the quantization process can be formalized as follows.

$$\bar{v} = \frac{v - \min v}{\max v - \min v} \times (2^n - 1).$$

(4)

In practice, only the indices $I \in \{0, \cdots, n\}$ for each element are communicated. Therefore, we map the data to the indices before conducting compression. Moreover, we consider sparsification, which selectively transmits a subset of gradients based on the specified proportion. When considering sparsification, it is important to note that the gradients remain as 32-bit floating points. For this experiment, we investigate quantization levels of 16, 8, and 1 bit (i.e., SignSGD [Bernstein et al., 2018]), and sparsification levels of 50%, 25%, and 10%.

We present a compatibility analysis in Fig. 4. The results indicate that integrating lossless compression techniques such as LZMA and LM-GC enhances compression rates beyond plain lossy compression. However, LZMA shows limited improvement across all settings, particularly with sparsification. In contrast, our method consistently delivers improvements across all settings, achieving notable compression rates in addition to lossy compression. These findings underscore the potential of LLMs as a prior for gradient compression, even with the incorporation of additional compression schemes, suggesting a promising new research direction in leveraging LLMs for compression.

## 6 Discussion and Limitation

**Throughput.** Despite the promising performance and generalizability, the throughput of LM-AC can be further optimized. Currently, our approach requires approximately 4 hours to compress just 28 MB. This bottleneck arises primarily from two components: LLMs and arithmetic coding. For LLMs, performance can be accelerated through techniques such as quantization [Frantar et al., 2023], faster attention mechanisms [Dao et al., 2022], KV cache Hooper et al. [2024], and model pruning Ma et al. [2023]. Looking ahead, one could explore distilling language models [Hsieh et al., 2023], as many functionalities may not be necessary during compression. Additionally, our implementation is significantly hindered by arithmetic coding and CPU limitations. Adopting a more efficient implementation, such as pure C++ programs, or utilizing CPUs with superior single-thread processing speeds could effectively mitigate these constraints.

**Broader impact.** Our work highlights the potential of leveraging pre-trained LLMs as priors for gradients. Immediately, this offers an advanced tool for gradient compression that reduces resource demands in federated and distributed learning environments. Over time, these priors could be utilized for gradient denoising, enhancing differential privacy training, or identifying adversarial gradients concealed within federated learning clients. However, this approach may also enable more subtle adversarial gradients, guided by these stronger priors.

## 7 Conclusion

We presented LM-GC, the first lossless gradient compressor that integrates arithmetic coding with LLMs as prior models for gradients. Our experiments show that pre-trained zero-shot LLMs are highly effective as gradient priors, setting a new state-of-the-art for gradient compression. Additionally, our findings indicate that the precise serialization of gradients substantially improves the reasoning abilities of LLMs and significantly impacts compression performance, warranting further exploration. The versatility of LM-GC sets the stage for developing more sophisticated gradient compression methods that directly incorporate LLMs. Overall, while our results in zero-shot settings are promising, the potential of expanding this approach to include few-shot learning, prompt engineering, and optimization of throughput efficiency remains open for further exploration.

## Acknowledgements

This work is partially funded by Medizininformatik-Plattform "Privatsphären-schutzende Analytik in der Medizin" (PrivateAIM), grant No. 01ZZ2316G, and Bundesministeriums fur Bildung und Forschung (PriSyn), grant No. 16KISAO29K. The work is also supported by ELSA – European Lighthouse on Secure and Safe AI funded by the European Union under grant agreement No. 101070617. Moreover, The computation resources used in this work are supported by the Helmholtz Association's Initiative and Networking Fund on the HAICORE@FZJ partition. Views and opinions expressed are, however, those of the authors only and do not necessarily reflect those of the European Union or European Commission. Neither the European Union nor the European Commission can be held responsible for them.

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

# Appendix

## A    Run Length Encoding

| RLE (bits) | RLE ($H_n$) | RLE (ISO) | LM-GC ($H_s$) |
|---|---|---|---|
| 450.28±0.3 | 278.08±0.2 | 198.57±0.0 | **71.90±0.0** |

Table 4: Run length encoding results of gradients collected from ConvNets trained on TinyImageNet.

We additionally compare our method to run-length encoding (RLE). RLE compresses data by counting the consecutive symbols and replaces the original data with a series of (`counts, symbol`) tuples. It serves as a simple adaptive compression codec without knowing data characteristics. The experiment extends from Table 3, compressing gradients collected during training a ConvNet on TinyImageNet. We consider three types of dictionaries: binary, hexadecimal without separators ($H_n$, Table 1), and iso-8859-1 (extended ASCII to handle negative numbers). These methods use 1, 4, and 8 bits to represent symbols and always use 8 bits for counting. Note that this setting is favorable to RLE since gradient lengths can easily exceed 256 (8 bits).

The results are presented in Table 4. While different codebooks improve the efficacy of RLE, RLE failed to compress the data and even increase the data size. On the other hand, our method clearly outperforms RLE, indicating that simple adaptive priors are ineffective for gradients.

