# OpenReview forum: "Language Models as Zero-shot Lossless Gradient Compressors: Towards General Neural Parameter Prior Models"
_NeurIPS.cc/2024/Conference — NeurIPS 2024 poster_

### Official Review · Reviewer_UCGG · 2024-07-11

**Soundness:** 4
**Presentation:** 3
**Contribution:** 3
**Rating:** 6
**Confidence:** 3

**Summary:**

The paper explores the novel use of large language models (LLMs) as gradient priors in a zero-shot setting. The primary focus is on lossless gradient compression, which is essential for distributed learning applications. The paper introduces LM-GC, a method that integrates pre-trained LLMs with arithmetic coding to transform gradients into a text-like format that LLMs can process efficiently. This approach significantly improves compression rates, surpassing existing state-of-the-art lossless compression methods by 10% to 21%. The method also shows compatibility with lossy compression techniques like quantization and sparsification.

**Strengths:**

1. The paper leverages pre-trained LLMs in an unconventional way, using them as priors for gradient compression, demonstrating creativity and novelty.
2. The proposed method, LM-GC, consistently outperforms traditional compression methods (PNG, FLAC, GZIP, LZMA) across various datasets and architectures.
3. Extensive experiments validate the effectiveness of LM-GC, including ablation studies and compatibility analyses with other compression techniques.
4. The paper discusses potential applications of LLMs in gradient denoising, differential privacy training, and adversarial gradient detection, highlighting broader impacts and future research directions.

**Weaknesses:**

1. The method incurs significant computational overhead, with compression times being relatively high.
2. The approach involves multiple hyper-parameters, such as context window size and byte grouping, which require careful tuning for optimal performance.
3. The integration of LLMs with arithmetic coding and the serialization process might be complex to implement and replicate.

**Questions:**

1. How well does LM-GC generalize to tasks and models not covered in the experiments? Are there specific scenarios where this approach might not be as effective?
2. How scalable is the proposed method for extremely large models or distributed training setups? Are there any known limitations in this regard?
3. How sensitive is the performance of LM-GC to different serialization strategies? Could there be more optimal serialization methods not explored in this paper?
4. What specific techniques or optimizations could be employed to reduce the computational overhead and improve the throughput of LM-GC?

**Limitations:**

see the questions.

---

> ### Author Rebuttal · Authors · 2024-08-05
>
> We thank the reviewer for their time and detailed feedback. We are also encouraged by the recognition of our novelty and creativity. We now address the comments and share our vision for further efficiency improvement below.
>
> ----
> ***Computation overhead and potential mitigation***
>
> We have discussed several potential strategies in the general response. In summary, there are three possible directions: advanced implementation, efficient LLMs, and hardware acceleration.
>
> - **Advanced Implementation**: This improves our current research-oriented single-thread Python implementation.
> - **Efficient LLMs**: This is an ongoing research direction aimed at developing more lightweight LLMs.
> - **Hardware Acceleration**: This leverages the zero-shot property and is intended for production use.
>
> ----
> ***Hyper-parameter selection***
>
> Interestingly, our findings suggest that most hyperparameters align with human intuitions. For example, 4 bytes per group resemble floating point structures and yield the best performance, while hexadecimal encoding outperforms iso-8859-1 due to the more common characters used. Other parameters generally adhere to the scaling law, where larger models or context windows result in better performance, creating a trade-off between performance and available resources. These factors significantly reduce the complexity of hyperparameter search.
>
> ----
> ***Implementation efforts***
>
> Our implementation can be easily reproduced using common open-sourced packages like PyTorch, Huggingface, and TorchAC, while serialization can be done using pure Python string operations. We will release the source code upon publication.
>
> ----
> ***Generalizability***
>
> Our approach performs well on common benchmarks, including MNIST, CIFAR-10, and TinyImageNet, and on architectures like ConvNet, VGG, ResNet, and ViT for image classification. Further analysis may be needed for other tasks, such as generative models, particularly if different implicit biases are present. To the best of our knowledge, this remains an open question, and we will leave this analysis for future work.
>
> ----
> ***How scalable is the proposed method for extremely large models or distributed training setups?***
>
> We currently target scenarios like federated/distributed learning, where gradients are compressed before sending to the server. We tested on models up to 6M parameters, a common model size for edge-device scenarios. Though the typical parameter size of LoRA fine-tuning is also around 6M, further analysis would be required on those extremely large models.
>
> ----
> ***How sensitive is the performance of LM-GC to different serialization strategies?***
>
> Table 1 shows that LLMs are sensitive to serialization strategies. For instance, hexadecimal numbers are up to 70% better than the extended ASCII set (iso-8859-1), and separators play a critical role in the reasoning ability of LLMs (40% difference for LLAMA 2).
>
> ----
> ***Could there be more optimal serialization methods not explored in this paper?***
>
> Yes, there may exist better strategies. This could be similar to the existing multi-modal LLM problems, where the key challenge is to project a new modality to the token space. While we propose an easy-to-go serialization method, understanding how and why LLMs can understand gradients deserves further research.

---

### Official Review · Reviewer_Nmkh · 2024-07-12

**Soundness:** 4
**Presentation:** 4
**Contribution:** 4
**Rating:** 7
**Confidence:** 2

**Summary:**

This is an interesting paper that tries to solve an important problem using large language models. Authors propose to use LLMs to compress the gradients with no number representation loss. They explain their method clearly by marrying LLMs with arithmetic coding.

**Strengths:**

This paper opens new views to LLMs as a tool in compression and coding. Gradient compression as stated in the paper is largely ignored. They prove LLMs can be used effectively as a prior for gradients in a zero-shot setting.

**Weaknesses:**

I would suggest that authors think about a proper theory to be added in the revised paper, or in the appendix. This combination is new and I suggest that some theoretical properties be developed in addition to experiments.

**Questions:**

Heavy tail priors are observed in training of deep models. Bring more context about number formats when the inherent distribution does not have expectation or variance, for instance Student-t distribution with 1 or 2 degrees of freedom.

**Limitations:**

When LLM is not a good tool for gradient compression? Under what circumstances they fail to present the prior.

---

> ### Author Rebuttal · Authors · 2024-08-05
>
> We thank the reviewer for their time and constructive feedback. We especially appreciate the recognition of our novelty and our efforts to address this timely question. We are pleased to share additional insights below.
>
> ----
> ***Theoretical properties***
>
> Thanks for the suggestion. We agree that theoretical analysis could be insightful. The fundamental theories behind entropy coding have been well-studied in the existing literature. On the other hand, understanding the behavior of LLMs is a non-trivial ongoing research direction. We will leave it to future work at this point.
>
> ----
> ***Clarification on the question***
>
> We are not entirely clear on the question. We would be more than happy to address it during the discussion phase if the reviewer could kindly provide further clarification.
>
> ----
> ***When is LLM not a good tool for gradient compression?***
>
> The main limitation of our approach is its high computational resource demand, making it less suitable for weaker devices such as IoT devices or real-time systems. However, this constraint could be significantly improved with the techniques discussed in our general response, enhancing scalability. Furthermore, our method does not require any training, making it especially well-suited for hardware acceleration in production deployments.
>
> ----
> ***Under what circumstances do they fail to present the prior?***
>
> Our approach performs well on common benchmarks, including MNIST, CIFAR-10, and TinyImageNet, and on architectures like ConvNet, VGG, ResNet, and ViT for image classification. Further analysis may be needed for other tasks, such as generative models, particularly if different implicit biases are present. To the best of our knowledge, this remains an open question, and we will leave this analysis for future work.

---

### Official Review · Reviewer_KkqK · 2024-07-12

**Soundness:** 1
**Presentation:** 3
**Contribution:** 2
**Rating:** 5
**Confidence:** 3

**Summary:**

The paper proposes LLMs as compressors for gradients of a neural network. The target usecase is a distributed learning setting, where the gradient updates need to be compressed before being shared.

The gradients are converted to hexadecimal representation and the LLM's outputs are used in an arithmetic coder to generate the compressed representation.

Post rebuttal edit: Improved rating 3->5

**Strengths:**

The idea is certainly novel. I do not know of prior works using LLMs to compress numerical data.

It works better than the baseline methods as demonstrated in the experiments across multiple model architectures and image datasets.

The LLM's don't require any dataset-specific training therefore they can serve as a general prior for floating-point datastreams.

**Weaknesses:**

The method lacks comparison to proper baseline methods.
> Line 195: "We compare our method to state-of-the-art lossless compression techniques that originally targeted different data types."

There needs to be justification why there isn't a comparison to a baseline method that is targeted at a stream of floating point values (or even specialized to gradients). I don't see significant advantages for using LLMs (or methods designed for image/audio compression) over a method specialized the task.
* There should to be a comparison to a simple adaptive entropy coder such as run-length coding or LZW.
* There should to be a comparison to a static codebook + arithmetic coding.
Here are a few other works that specialize in compressing floating point values:
* https://computing.llnl.gov/projects/fpzip Fast and Efficient Compression of Floating-Point Data
* https://ieeexplore.ieee.org/document/1607248 Fast lossless compression of scientific floating-point data

Besides, it is also clearly impractical to use LLMs for this task due to their computational costs. The LLMs use excessively more compute than the bandwidth savings could justify.

**Questions:**

Figure 4: What does 'plain' compression entail here? Is it using a codebook with arithmetic coding?

**Limitations:**

The solution proposed is extremely resource-intensive for the given task, which may have societal impacts.

---

> ### Author Rebuttal · Authors · 2024-08-05
>
> We thank the reviewer for their time and insightful feedback. We hope our response addresses the concerns and conveys the broader vision of our work.
>
> We kindly emphasize that our primary contribution is demonstrating **the potential of pre-trained zero-shot large language models (LLMs) as effective gradient priors through the lens of lossless compression**, a point also supported by R-UCGG and R-Br5V. As motivated in L21-23, the lack of proper prior models has been a critical barrier for applications such as denoising or upsampling in the domain of gradients. Our work is the first to demonstrate this potential and explore its application in lossless gradient compression. We believe our findings will inspire further research beyond just arithmetic coding. The potential impact also outweighs the temporal computation overhead as we have discussed in the general response.
>
> We now address the concerns regarding the lossless gradient compression task below.
>
> ----
>
> ***Justification of using LLMs.***
>
> We first emphasize that traditional baselines struggle to compress gradients effectively in complex scenarios. As illustrated in Tables 2 and 3, our method exceeds the best baseline by 20% on ViT and 21% on TinyImageNet. This highlights the difficulty of modeling complex gradients and justifies the usage of our LLM prior. The following experiments recommended by the reviewer further stress the point.
>
> ----
> ***Missing baselines.***
>
> We have compared our method to LZMA, one of the most robust **general-purpose** compression methods, commonly used in the 7z compression format. Additionally, it is important to note that **none of the suggested baselines are designed for gradients or high-dimensional data**. However, to further highlight our contribution, we have conducted additional experiments as follows.
>
> ----
> **Comparison to simple adaptive coding**
>
> We additionally compare our method to run-length encoding (RLE). This experiment extends from Table 3, compressing gradients collected during training a ConvNet on TinyImageNet. For RLE, we consider three types of dictionaries: binary, hexadecimal ($H_n$, Table 1), and iso-8859-1 (extended ASCII to handle negative numbers). These methods use 1, 4, and 8 bits to represent symbols and always use 8 bits for counting. Note that this setting is favorable to RLE since gradient lengths can easily exceed 256 (8 bits).
>
> The results are presented in the following table. RLE failed to compress the data even with different codebooks, and our method clearly outperforms RLE, indicating that simple adaptive priors are ineffective for gradients.
>
> | RLE (bits) | RLE ($H_n$) |  RLE (ISO) | LM-GC ($H_s$) |
> |:----------:|:-----------:|:----------:|:-------------:|
> | 450.28±0.3 |  278.08±0.2 | 198.57±0.0 | **71.90±0.0** |
> ----
> ***Comparison to methods dedicated to floating-point compression***
>
> While we acknowledge that the suggested baselines are intended for scientific floating points (1D to 4D, which are relatively low-dimensional compared to gradients [1]), we have extended the comparison in Table 2 and 3 to include floating-point-specific compression. Specifically, we compare our method to FPZIP [1], as we could not find any available implementation of [2]. The results, shown in Table A1 and A2 in the attached PDF, indicate that FPZIP performs comparably to LZMA. On the other hand, our LM-GC outperforms FPZIP by up to 17.2% across three datasets and around 20% across four architectures. This highlights the challenge of using heuristically designed priors and demonstrates the effectiveness of the LLM priors adopted in our method.
>
> ----
> ***Practicability of LM-GC***
>
> As previously discussed, the effectiveness of LLMs in handling complex gradients justifies their usage. We acknowledge that LM-GC requires more resources than traditional codecs. However, as detailed in the general response, significant acceleration can be achieved by implementing C++ and multi-threading. Additionally, ongoing research into more efficient LLMs suggests a promising future for further improvements. Notably, since our method does not require training, hardware can frequently achieve substantial acceleration for inference.
>
> ----
> ***What does 'plain' compression entail here? Is it using a codebook with arithmetic coding?***
>
> Plain means without any lossless compression. We will revise the text in the next version to avoid confusion.
>
> ----
> We hope our response has taken care of your concerns and encouraged you to reconsider the score. We would be happy to answer any other questions or concerns during the discussion phase.
>
> [1] Fast and Efficient Compression of Floating-Point Data https://computing.llnl.gov/projects/fpzip
> [2] Fast lossless compression of scientific floating-point data https://ieeexplore.ieee.org/document/1607248

---

> > ### Comment · Reviewer_KkqK · 2024-08-12
> > **Thank you for your reply.**
> >
> > Thank you for your reply and for providing further experimental measurements. It helped to alleviate my concerns regarding the performance improvement.
> >
> > In my opinion, it is very interesting to see how well LLMs perform compared to traditional coding methods. Showing that there is a significant gap will help to inform future gradient compression algorithms and provide a strong baseline. We can study LLMs to figure out why they perform so well and where the current algorithms are lacking.
> >
> > I am still convinced that the use of LLMs for this purpose is impractical, and will remain that way for the foreseeable future, but the paper's findings have value for the research community.

---

> > > ### Author Response · Authors · 2024-08-13
> > >
> > > Thank you for your prompt response and re-evaluation. We will definitely include the discussion and the new baselines in the next version.

---

### Official Review · Reviewer_Br5V · 2024-07-13

**Soundness:** 3
**Presentation:** 2
**Contribution:** 3
**Rating:** 5
**Confidence:** 3

**Summary:**

This paper studies the potential of LLMs to act as gradient priors in zero-shot settings. The property is evaluated on lossless gradient compression. The proposed method is able to surpass existing compression methods and improve compression rate by 10% to 21% across various datasets and architectures.

**Strengths:**

1. This paper explores the potential of LLMs as gradient priors through the lens of lossless gradient compression. Given the massive model size and training datasets, LLMs are encoded with many capacities that can be used as tools in domains that are out of the text domain, such as optimizers https://arxiv.org/abs/2309.03409, and compressors https://arxiv.org/abs/2309.10668. This paper is one of these, which itself is fun and interesting.

2. The effectiveness of the methodology design such as Serialization and Compression is backed up with promising results.

**Weaknesses:**

1. While this paper itself is fun, one concern remains what are the benefits of using LLM as a compressor if traditional approaches can sufficiently solve the problem? It is better to report the energy costs of LLMs and traditional baselines such as PNG, FLAC, LZMA, and GZIP to allow readers to have a full picture.

2. What are the potentials of this paper's finding? Whether the compressed gradient can be utilized in practice? for instance, how to explore this to train neural networks?

**Questions:**

please see the above weaknesses.

**Limitations:**

provided.

---

> ### Author Rebuttal · Authors · 2024-08-05
>
> We thank the reviewer for their time and feedback and are encouraged that our work was found interesting. We will now address the comments and share further vision of our work.
>
> ----
> ***The benefit of using LLMs over traditional approaches.***
>
> We highlight that traditional baselines struggle to compress gradients effectively in complex scenarios. As illustrated in Tables 2 and 3, our method exceeds the best baseline by 20% on ViT and 21% on TinyImageNet. This highlights the difficulty of modeling complex gradients and showcases the effectiveness of our LLM prior. Moreover, the baselines often fail to compress data adequately in some scenarios. For instance, LZMA, the best baseline, achieves only a 91% compression rate. These advantages demonstrate the benefits of our method, making it particularly favorable when communication cost or storage is the main system bottleneck.
>
> ----
> ***Energy consumption***
>
> We provide the energy consumption estimation in Figure A2 of the attached PDF. The consumption is estimated by the following equation:
>
> $
> E = (\text{CPU power consumption} \times \text{CPU utilization} + \text{GPU power consumption} \times \text{GPU utilization}) \times \text{run time}
> $
>
> Indeed, LM-GC currently consumes more energy. However, as discussed in the general response, we remain positive about future efficiency improvements. Moreover, in complex scenarios, the baselines often struggle to compress the data effectively (70.98% vs. 87.98% in Tables 2 and 71.90% vs. 91.6% in Table 3), which justifies our method's investment in computational resources.
>
> ----
> ***Can the compressed gradient be utilized in practice?***
>
> Yes. An immediate application is federated learning, where clients accumulate gradients locally and communicate periodically with the central server. One of the main bottlenecks is the communication cost as the number of clients and model size grow. Assuming both sender and receiver have access to the same LLM, LM-GC (optionally with lossy compression, as shown in Figure 4) provides superior compression rates on gradients over the existing coding schemes (Table 1-3) and thus enables better scalability.
>
> ----
> ***Broader impact of our findings***
>
> In addition to LM-GC, our work highlights the potential of using large language models (LLMs) as gradient priors, which we find exciting and believe will inspire further applications. We propose several possible directions for exploration, which could be investigated in fully zero-shot, parameter-efficient fine-tuning (PEFT), or prompting settings, given the strong starting point provided by zero-shot LLMs.
>
> - **Inspiration from existing works**: With a robust prior $p(t_k|\texttt{BOS}, t_{<k})$ over gradient tokens, it is possible to draw inspiration from existing work of different modalities. For example, one could utilize the prior model to de-noise gradients by checking the likelihood, akin to image denoising with generative priors, or explore gradient restoration from quantized low bits to higher bits, similar to image super-resolution. In the latter scenario, the lower bits provide partial information or context for the LLMs.
>
> - **Advanced lossy compression**: Most of the existing lossy compression in federated learning assumes no prior knowledge exists between servers and clients or leverages heuristically designed protocols. In contrast, one can assume both sides have access to a fixed LLM. When certain tokens consistently exhibit high probability, they can be sampled directly from the LLMs, allowing more bits to be allocated to other tokens.
>
> - **Security**: By modeling benign gradients, potentially through PEFT or prompting, one could develop a gradient-based method to guard against backdoor attacks, akin to image anomaly detection using generative adversarial networks (GANs).

---

### Author Rebuttal · Authors · 2024-08-05

# General Response
\
We thank all reviewers for their time and constructive feedback. It is encouraging to hear that our work has been found both fun and interesting (R-Br5V) as well as creative (R-UCGG). We particularly appreciate the recognition from all reviewers of the novelty and effectiveness of our method across various datasets and architectures (R-Br5V, R-KkqK, R-Nmkh, R-UCGG). We will begin by recapping the importance of our work and then address the comments in each thread, respectively.

----

### ***Importance of our work***

Our study is the first to investigate the applications of *general-purpose* LLMs in the field of *gradients*. In addition to traditional NLP tasks, we show for the first time that LLMs can act as a robust gradient prior model and comprehend gradients *without requiring fine-tuning or demonstrations*. This discovery will expand the potential uses of general-purpose LLMs and inspire further applications involving gradients, as discussed in L21-23. For instance, our LM-GC, an immediate application of LLM priors in lossless gradient compression, presents superior performance, especially in complex scenarios that traditional methods struggle to model. Despite the room for improvement in our implementation, our research opens up an entirely new direction that has never been explored. We are confident that our contribution outweighs these temporary computational burdens.

With that being said, we have noticed several potential strategies for further efficiency improvement. Many of them are still ongoing research problems. We anticipate that both the field and our work will mutually benefit in the near future.

- **Advanced Implementation**: Our current implementation is designed for research purposes and is a single-thread Python program that processes each window of gradients sequentially. Given that the average GPU utilization is around 10% and each window is independent, the implementation can be significantly improved by utilizing multi-threading and C++. This could potentially yield a 10 times improvement in speed.

- **Efficient LLMs**: Accelerating LLMs is a popular research problem. Techniques such as quantization [1], KV caches [2], flash attention [3], and pruning [4] are already being explored. One promising but underexplored direction is distillation [5]. While pre-trained language models are powerful, we do not need all of their functionalities for our task. Extracting the desired functions (e.g., gradient priors) from LLMs might be a significant step toward efficient LM-GC.

- **Hardware Acceleration**: LM-GC does not require any training. Inference of networks can often be significantly accelerated by AI-specific hardware such as NPUs, TPUs, or other application-specific integration circuits (ASICs).

[1] Lin, Ji, et al. "AWQ: Activation-aware Weight Quantization for On-Device LLM Compression and Acceleration." Proceedings of Machine Learning and Systems 6 (2024): 87-100.\
[2] Hooper, Coleman, et al. "Kvquant: Towards 10 million context length llm inference with kv cache quantization." arXiv preprint arXiv:2401.18079 (2024).\
[3] Shah, Jay, et al. "Flashattention-3: Fast and accurate attention with asynchrony and low-precision." arXiv preprint arXiv:2407.08608 (2024).\
[4] Ma, Xinyin, Gongfan Fang, and Xinchao Wang. "Llm-pruner: On the structural pruning of large language models." Advances in neural information processing systems 36 (2023): 21702-21720.\
[5] Hinton, Geoffrey, Oriol Vinyals, and Jeff Dean. "Distilling the knowledge in a neural network." arXiv preprint arXiv:1503.02531 (2015).

---

### Decision · Program_Chairs · 2024-09-25

**Decision:**

Accept (poster)

**Comment:**

This paper uses LLMs as priors for arithmetic coding, showing that LLMs can be used to compress gradients of neural networks. Reviewers unanimously applauded the originality and ingenuity of the method, and I agree. By showing this unexpected use of LLMs, I believe this work is likely to lead to follow-up research.

Reviewers also raised the concern of computational cost as a limitation, which was discussed during the rebuttal. I ask the authors to please include some of the discussion and additional results from the rebuttal into the final version of the manuscript so as to make it clear throughout the manuscript that this is a limitation of the current incarnation of the method.